# Modeling toes contributes to realistic stance knee mechanics in three-dimensional predictive simulations of walking

**Antoine Falisse** [1,2] *, **Maarten Afschrift** [3,4], **Friedl De Groote** [2]

**1** Department of Bioengineering, Stanford University, Stanford, California, United States of America,
**2** Department of Movement Sciences, KU Leuven, Leuven, Belgium, **3** Department of Mechanical Engineering, Robotics Core Lab of Flanders Make, KU Leuven, Leuven, Belgium, **4** Department of Human Movement Sciences, Vrije Universiteit Amsterdam, Amsterdam, The Netherlands

* afalisse@stanford.edu

**Data Availability Statement:** All data, code, and materials used in this study are available at https://simtk.org/projects/predictsim_mtp.

## Abstract

Physics-based predictive simulations have been shown to capture many salient features of human walking. Yet they often fail to produce realistic stance knee and ankle mechanics. While the influence of the performance criterion on the predicted walking pattern has been previously studied, the influence of musculoskeletal mechanics has been less explored. Here, we investigated the influence of two mechanical assumptions on the predicted walking pattern: the complexity of the foot model and the stiffness of the Achilles tendon. We found, through three-dimensional muscle-driven predictive simulations of walking, that modeling the toes, and thus using two-segment instead of single-segment foot models, contributed to robustly eliciting physiological stance knee flexion angles, knee extension torques, and knee extensor activity. Modeling toes also slightly decreased the first vertical ground reaction force peak, increasing its agreement with experimental data, and improved stance ankle kinetics. It nevertheless slightly worsened predictions of ankle kinematics. Decreasing Achilles tendon stiffness improved the realism of ankle kinematics, but there remain large discrepancies with experimental data. Overall, this simulation study shows that not only the performance criterion but also mechanical assumptions affect predictive simulations of walking. Improving the realism of predictive simulations is required for their application in clinical contexts. Here, we suggest that using more complex foot models might contribute to such realism.

## Introduction

Predictive simulations have advanced our understanding of gait neuromechanics but also revealed gaps in our current knowledge [1]. Motivated by the observation that humans select temporospatial walking features (e.g., step frequency [2] and stride length [3]) that minimize metabolic energy per distance traveled, predictive simulations of walking commonly rely on the assumption of performance optimization. *De novo* walking patterns can therefore be generated by solving for muscle controls that optimize a performance criterion based on a model

**Funding:** This work was supported by KU Leuven (https://www.kuleuven.be) through internal funds C24M/19/064 to F.D.G. The funders had no role in study design, data collection and analysis, decision to publish, or preparation of the manuscript.

**Competing interests:** The authors have declared that no competing interests exist.

of musculoskeletal dynamics with task-level constraints such as walking speed and periodicity. This approach has produced walking patterns that capture many salient features of human gait, indicating that performance optimization and musculoskeletal dynamics are important determinants of gait mechanics. Yet several gait features have remained hard to capture. Predicting knee flexion during stance along with physiologically plausible knee extension torques and activation patterns of the knee extensors (vasti) has been challenging [4]. Another gait feature that has been hard to simulate, but has received less attention, is ankle kinematics during stance. Understanding why predictive simulations do not capture those gait features well will advance our knowledge of gait neuromechanics and is key to improving the realism of predictive simulations, which in turn is important to enable the use of simulations for designing gait interventions in clinical settings. While the role of the performance criterion has been explored, it remains unclear how musculoskeletal modeling assumptions contribute to the lack of stance knee flexion and poor ankle kinematics in simulations of walking.

The performance criterion underlying predictive simulations of walking has been shown to influence the degree of knee flexion during stance in two-dimensional (2D) walking simulations. Yet it has been difficult to identify a criterion that robustly generates such flexed knee patterns. Ackermann and van den Bogert [5] found that stance knee flexion was influenced by the cost function in muscle-driven 2D simulations of walking. They compared energy-like cost functions, sum of muscle-volume-scaled activations to the power 1–4, and fatigue-like cost functions, sum of activations to the power 2–10, and found that energy-like cost functions resulted in straight stance knee patterns with low knee extensor activations, whereas fatigue-like cost functions resulted in more realistic knee flexion angles and knee extensor activations during stance. However, these observations might not be robust against other modeling choices as they were confirmed by some (e.g., [6]) but not all future predictive simulation studies of walking. For example, our recent 2D muscle-driven simulations of walking based on a similarly complex model lacked stance knee flexion although we solved for muscle controls that minimized a fatigue-like cost function (i.e., muscle activations cubed) [7]. Further, Koelewijn et al. [8] obtained realistic stance knee flexion, nevertheless in the absence of knee extension torques and knee extensor activations, when optimizing for metabolic energy.

Finding a performance criterion that produces realistic stance knee flexion has also been challenging when using more complex three-dimensional (3D) musculoskeletal models. Anderson and Pandy [9] simulated a 3D walking motion by solving for controls that minimized metabolic energy. They enforced knee flexion during stance by imposing joint kinematics at the initial time instant of the simulation, chosen to be mid-stance, to match experimental data. Miller [10] found that metabolic energy models influenced the simulated walking pattern, with most models predicting some, albeit less than experimentally observed, knee flexion during stance. All simulated walking patterns underestimated ankle plantarflexion in late stance. In recent work, we found that minimizing a cost function that penalized both metabolic energy and muscle activity yielded more realistic walking patterns than minimizing a cost function that penalized either metabolic energy or muscle activity [4]. Yet our simulations underestimated stance knee flexion, knee extension torques, and knee extensor activations, and produced poor stance ankle kinematics. Hence, control objectives other than reducing metabolic cost or fatigue as well as musculoskeletal modeling assumptions might also play a role in shaping stance knee and ankle mechanics.

Here, we demonstrate that musculoskeletal mechanics influences knee flexion during stance. We first focused on the foot, which is commonly modeled as a single-segment rigid body despite the clear extension of the toe joints during walking. We found, through 3D simulations of walking, that a two-segment foot model that allowed movement between the toes and the rest of the foot led to more accurate predictions of knee kinematics, knee kinetics, and

ankle kinetics than a single-segment foot model. Yet modeling toe joints slightly worsened predictions of ankle kinematics. We therefore also explored the influence of the Achilles tendon stiffness on the simulated walking pattern. First, Achilles tendon compliance has been suggested to improve gait efficiency both by allowing for storage and release of energy throughout the gait cycle and through its influence on the operating length and velocity of the plantarflexors [11–15]. Since it is the interaction between ankle kinematics and Achilles tendon properties that determines plantarflexor operating length and velocity, and thereby efficiency, adjusting the Achilles tendon stiffness in our simulations that optimize efficiency might influence predicted ankle kinematics. Second, we previously found that reducing Achilles tendon stiffness in our models resulted in inverse dynamic estimates of gastrocnemius fiber length trajectories that were in closer agreement with fiber length trajectories measured using ultrasonography during walking [16] and running [17]. In inverse dynamic simulations of walking, we obtained the best agreement between simulated and measured fiber lengths when decreasing Achilles tendon stiffness by 60%. Finally, since the gastrocnemii, attached to the Achilles tendon, cross both knee and ankle joints, we expected changes in Achilles tendon stiffness to affect both joints and therefore to potentially improve our kinematic predictions at both levels. We found that decreasing the Achilles tendon stiffness by up to 60% slightly improved predictions of ankle kinematics.

## Methods

The modeling and simulation workflow is described in detail in previous work [4, 7].

### Musculoskeletal model

We used an OpenSim musculoskeletal model with 31 degrees of freedom (DoFs) [18, 19] (pelvis-to-ground: 6 DoFs, hip: 3 DoFs, knee: 1 DoF, ankle: 1 DoF, subtalar: 1 DoF, metatarsophalangeal-toe: 1 DoF, lumbar: 3 DoFs, shoulder: 3 DoFs, and elbow: 1 DoF), 92 muscles actuating the lower limb and lumbar joints, eight ideal torque motors actuating the shoulder and elbow joints, and six contact spheres per foot. To increase computational speed, we fixed the moving knee flexion axis of the generic model to its anatomical reference position [20]. We added passive stiffness (exponential) and damping (linear) to the lower limb and lumbar joints, henceforth referred to as passive torques, to model ligaments and other passive structures [9].

We adjusted the orientation of the toe joint coordinate frame of the generic OpenSim model to be aligned with that of its parent frame, namely the calcaneus. We empirically found that an orientation of the toe joint axis perpendicular to the sagittal plane in the anatomical position better reproduced the movement of the toes during walking as compared to the generic orientation of the toe joint axis through the base of the metatarsals. We confirmed through preliminary simulations that using a toe joint axis that was perpendicular to the sagittal plane resulted in simulated walking patterns that were in better agreement with experimental data than using the generic toe joint axis. The toe joints had no active actuation. On top of the aforementioned passive torques, we added a linear rotational spring with a stiffness of 25 Nm rad$^{-1}$ [21] and a damper with a damping coefficient of 2 Nm s rad$^{-1}$ for the toe joints. We selected the damping value from a sensitivity analysis (\ Appendix). We used higher damping for the torque-driven toe joints as compared to the muscle-driven joints for which there is also damping at the muscle level.

We used Raasch's model [22, 23] to describe muscle excitation-activation coupling and a Hill-type muscle model [24, 25] to describe muscle-tendon interaction and the dependence of muscle force on fiber length and velocity. We modeled skeletal motion with Newtonian rigid

body dynamics and smooth approximations of compliant Hunt-Crossley foot-ground contacts [4, 19, 26, 27]. We described the dynamics of the ideal torque motors using a linear first-order approximation of a time delay. To increase computational speed, we defined muscle-tendon lengths, velocities, and moment arms as a polynomial function of joint positions and velocities. More details about the musculoskeletal model can be found in previous work [4].

## Experimental data

We used experimental data from a single subject averaged over 10 gait cycles for comparison with simulation outcomes as well as to provide some of the bounds and initial guesses of the predictive simulations [4]. We collected data (marker coordinates, ground reaction forces, and electromyography) from a healthy female adult (mass: 62 kg, height: 1.70 m). Marker coordinates were recorded (100 Hz) using a ten-camera motion capture system (Vicon, Oxford, UK), ground reactions forces were recorded (1000 Hz) using two force plates (AMTI, Watertown, USA), and electromyography was recorded (1000 Hz) using a wireless acquisition system (ZeroWire EMG Aurion, Milano, Italy) from 24 lower limb muscles (hamstrings lateralis and medialis [left and right], gluteus medius [left and right], adductor longus [left], tensor fasciae latae [left], vastus lateralis and medialis [left and right], rectus femoris [left and right], tibialis anterior [left and right], gastrocnemius lateralis [left and right] and medialis [left], soleus [left and right], and peroneus longus [left and right] and brevis [left]). The subject was instructed to walk over the ground at a self-selected speed. The average walking speed was $1.33 \pm 0.06$ m s$^{-1}$. We processed the experimental data with OpenSim 4.2 [19]. The musculoskeletal model was scaled to the subject's anthropometry based on marker information from a standing calibration trial. Joint kinematics were calculated based on marker coordinates through inverse kinematics. Joint kinetics were calculated based on joint kinematics and ground reaction forces. Electromyography was processed by band-pass filtering (second order dual-pass Butterworth filter, 20-400Hz), rectification, and low-pass filtering (second order dual-pass Butterworth filter, 10Hz), and used for evaluating simulated muscle activations.

## Predictive simulation framework

We formulated predictive simulations of walking as optimal control problems [4, 7]. We identified muscle excitations and gait cycle duration that minimized a cost function subject to constraints describing muscle and skeleton dynamics, imposing left-right symmetry, preventing limb collision, and prescribing gait speed (distance traveled by the pelvis divided by gait cycle duration).

Our cost function included metabolic energy rate, muscle activity, joint accelerations, passive torques, and excitations of the ideal torque motors at the arm joints, all terms squared:

$$J = \frac{1}{d} \int_0^{t_f} \left( w_1 \dot{E}^2 + w_2 a^2 + w_3 u_a^2 + w_4 T_p^2 + w_5 e_a^2 \right) dt, \tag{1}$$

where $d$ is distance traveled by the pelvis in the forward direction, $t_f$ is half gait cycle duration, $a$ are muscle activations, $u_a$ are accelerations of the lower limb and lumbar joint coordinates, $T_p$ are passive torques, $e_a$ are excitations of the ideal torque motors driving the shoulder and elbow joints, $t$ is time, and $w_{1-5}$ are weight factors. We modeled metabolic energy rate $\dot{E}$ using a smooth approximation of the phenomenological model described by Bhargava et al. [28], which describes metabolic energy rate as the sum of four terms: muscle activation, shortening, and maintenance heat rates, and mechanical work rate (details about model implementation in S1 Appendix). To avoid singular arcs [29], we appended a penalty function with the

remaining controls to the cost function. More details about the optimal control problem formulation can be found in previous work [4].

We formulated our problems in Python 3.7 using CasADi 3.5 [30], applied direct collocation using a third order Radau quadrature collocation scheme, used algorithmic differentiation to compute derivatives [7], and solved the resulting nonlinear programming problem with IPOPT [31]. We generated our simulations from two initial guesses and selected results with the lowest optimal cost values (i.e., integral of Eq 1 with optimal values). The first initial guess was a 'hot-start' based on experimental walking data, whereas the second initial guess was a 'cold-start' that consisted of a static, bilaterally symmetrical standing posture that translated forward over the simulation duration. Muscle states and controls were set to constant values for both initial guesses. We performed a convergence analysis by evaluating the sensitivity of the results to the number of mesh intervals (50, 75, 100, and 125 mesh intervals per half gait cycle) and to IPOPT convergence tolerance ($10^{-4}$ [default], $10^{-5}$, and $10^{-6}$). Increasing the mesh density had barely any influence on the results (i.e., state and control trajectories) (S1 Table), suggesting a good enough simulation accuracy. Yet it is worth noting that both initial guesses led to different solutions when using only 50 mesh intervals for models without toes, highlighting the need for different initial guesses. With higher mesh densities, both initial guesses led to very similar solutions. Overall, the number of iterations increased when increasing the mesh density, but with no clear pattern. At high mesh densities (100 and 125), the number of iterations was lower for the cold-start than for the hot-start. Tightening the convergence tolerance also had barely any influence on state and control trajectories (S2 Table), suggesting that the default tolerance of $10^{-4}$ is strict enough. On average, the number of iterations was more than twice and thrice as large for a convergence tolerance of $10^{-5}$ vs $10^{-4}$ and $10^{-6}$ vs $10^{-4}$, respectively. For all results of this study, we used 100 mesh intervals per half gait cycle (the dynamic equations were therefore satisfied at 300 collocation points) and a convergence tolerance of $10^{-4}$.

### Influence of the mass distribution and vertical location of the contact spheres

We investigated the effect of two main changes between the model used in the present study, henceforth referred to as new model, and the model used in our previous study [4], henceforth referred to as old model. The first change concerns the model mass distribution. In our previous study, we extended OpenSim's gait2392 model with arms without adjusting the mass of all other segments resulting in a generic (i.e., pre-scaling) model with a torso of 34.2 kg. Here, we started from the generic model proposed by Hamner et al. [18], which is also based on OpenSim's gait2392 model and is more commonly used. In Hamner's model the mass of the trunk was reduced when adding arms, resulting in a generic model with a torso of 26.8 kg. All other parameters, except inertia that scales with mass, were identical between both generic models. Since we scaled both models to the same subject mass, we obtained two scaled models with different mass distribution; the new model having a lighter torso, but heavier legs as compared to the old model. The second change concerns the vertical location of the contact spheres. In our previous study, we used the foot-ground contact configuration from Lin et al. [32]. However, our predictions had an offset in the vertical position of the pelvis when compared to experimental data. In the present study, we therefore manually adjusted the vertical location of the contact spheres, moving them higher up (about 1 cm) to reduce that offset. We evaluated the sensitivity of the predicted walking pattern to those two changes.

### Influence of the toe joints

We evaluated the influence of incorporating toe joints in the model on the predicted walking pattern. To this aim, we compared predictive simulations of walking produced with the

musculoskeletal model described above to simulations produced with the same model but with locked toe joints. In practice, for that model, we replaced the hinge toe joints by weld joints, thereby locking the toes in their neutral position. We performed this comparison for the old model (i.e., heavier torso and lighter legs) with low contact spheres as well as for the new model (i.e., lighter torso and heavier legs) with high contact spheres.

## Influence of the Achilles tendon stiffness

We evaluated the influence of the Achilles tendon stiffness on the predicted walking pattern. To this aim, we compared predictive simulations produced with Achilles tendon stiffness ranging from 30 to 100%, by increments of 10%, of the generic value ($k$ = 35 [25]). In more detail, we adjusted the variable $k$ for the triceps surae muscles (gastrocnemius lateralis and medialis, and soleus) in the following equation describing the tendon force-length relationship:

$$\tilde{l}_t = \frac{\log(5(\tilde{f}_t + 0.25 - s))}{k} + 0.995, \tag{2}$$

where $\tilde{l}_t$ is the normalized tendon length, $\tilde{f}_t$ is the normalized tendon force, and $s$ is a scaling parameter that shifts the tendon force-length curve as a function of the tendon stiffness $k$ to enforce that the tendon force produced when the normalized tendon length equals the tendon slack length is independent of the tendon stiffness. We used the new model with toes for those simulations.

For all results, we quantified the effect of the changes using root mean squared error (RMSE) of stance knee and ankle kinematics and kinetics with respect to mean experimental data. Furthermore, we quantified changes between models by expressing the difference in RMSE as a percentage of the signal's range during stance.

## Results

### Influence of the mass distribution and vertical location of the contact spheres

Adjusting the model mass distribution improved stance knee kinematics and kinetics (RMSE decreased from 10.2 to 8.6˚ [improvement of 4.6% when expressed as a percentage of the signal's range] and from 22.1 to 20.3 Nm [-2.7%], respectively), had barely any effect on ankle kinematics (no change in RMSE) and kinetics (RMSE increased from 15.0 to 15.4 Nm [0.4%]), and had a small effect on muscle activations (Fig 1). More specifically, the new model with a lighter torso and heavier legs produced slightly larger knee flexion angles, knee extension torques, and knee extensor (vasti) activations during stance. Moving the contact spheres higher up further improved stance knee kinematics and kinetics (RMSE decreased from 8.6 to 6.8˚ [-5.5%] and from 20.3 to 17.8 Nm [-3.8%], respectively) by producing higher knee flexion angles, knee extension torques, and knee extensor activations during stance. Ankle kinematics and kinetics slightly worsened with higher contact spheres (RMSE increased from 7.3 to 7.5˚ [0.8%] and from 15.4 to 17.0 [1.7%]). The mass distribution and contact location had little influence on the predicted metabolic cost of transport (between 3.89 and 3.93 J kg$^{-1}$ m$^{-1}$; Table 1), which is slightly above reported experimental measurements (3.35 ± 0.25 J kg$^{-1}$ m$^{-1}$ [10]).

### Influence of the toe joints

Predictive simulations with a model incorporating toe joints produced stance knee flexion angles, knee extension torques, and ankle flexion torques that better reproduced experimental

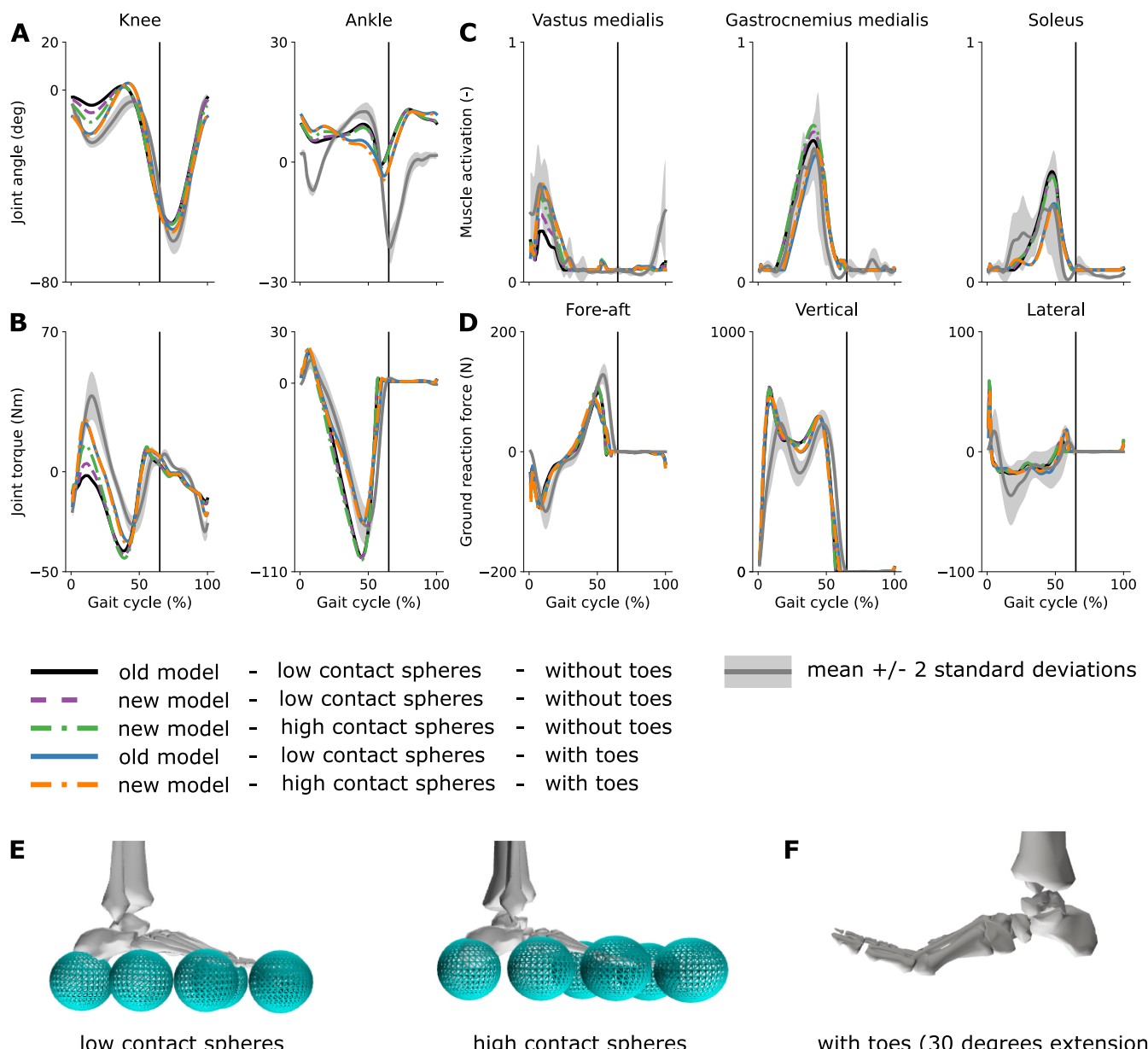

**Fig 1. Influence of modeling choices on predicted gait pattern.** Predicted knee and ankle kinematics (**A**) and kinetics (**B**), muscle activations (**C**), and ground reaction forces (**D**) with the different models. Experimental data (shaded areas) are shown as mean ± 2 standard deviations. The experimental electromyography data were normalized to peak activations of the new model with high contact spheres and with toes (dashdot orange curve). The vertical black lines indicate experimental stance to swing transition. The old and new models have the same mass but different mass distribution: the new model having a lighter torso, but heavier legs as compared to the old model. The difference in the vertical position of the contact spheres (about 1 cm) is illustrated in (**E**), and an example toe extension is depicted in (**F**). Results for all joints and muscles are shown in S1–S3 Figs.

data as compared to simulations produced with a model with locked toe joints (RMSE decreased from 6.8 to 5.3˚ [-4.5%], from 17.8 to 10.6 Nm [-11.2%], and from 17.0 to 6.2 Nm [-11.2%] for the stance knee kinematics, knee kinetics, and ankle kinetics, respectively) (Figs 1 and 2; results for all joints and muscles are shown in S1–S3 Figs; animations are shown in S1 and S2 Movies). The predicted stance ankle kinematics changed when incorporating toe joints but slightly worsened compared to experimental data (RMSE increased from 7.5 to 8.2˚ [2.4%]). Ground reaction forces produced with models with and without toe joints were

**Table 1. Influence of modeling choices on the metabolic cost of transport (COT).**

| | Without toe joints | | | With toe joints | | Reference (Miller [10]) |
|---|---|---|---|---|---|---|
| | Old model + low contact spheres | New model + low contact spheres | New model + high contact spheres | Old model + low contact spheres | New model + high contact spheres | |
| COT (J kg$^{-1}$ m$^{-1}$) | 3.89 | 3.90 | 3.93 | 4.22 | 4.22 | 3.35 ± 0.25 |

The new model has a lighter torso, but heavier legs as compared to the old model. The high contact spheres are about 1 cm higher than the low contact spheres.

qualitatively comparable, although incorporating toe joints decreased the first peak of the vertical forces (from 761 to 725 N), increasing its agreement with experimental data (620 N). In models with toe joints, the knee extensor (vasti) activations slightly increased during early stance, whereas the triceps surae (gastrocnemius and soleus) activations decreased during stance. The metabolic cost of transport was ~7% (8%) higher with the new (old) model with toe joints (4.22 J kg$^{-1}$ m$^{-1}$ for both old and new models) as compared to the new (old) model without toe joints (Table 1). The quadriceps (vasti and rectus femoris) mainly contributed to the increase in metabolic cost of transport (29% of the increase).

Interestingly the model adjustments detailed above (mass distribution and vertical location of the contact spheres) had barely any effect when incorporating toe joints in the model. In more detail, predictive simulations with the old model with a heavier torso and lighter legs, with low contact spheres, and with toe joints were almost identical as those produced with the

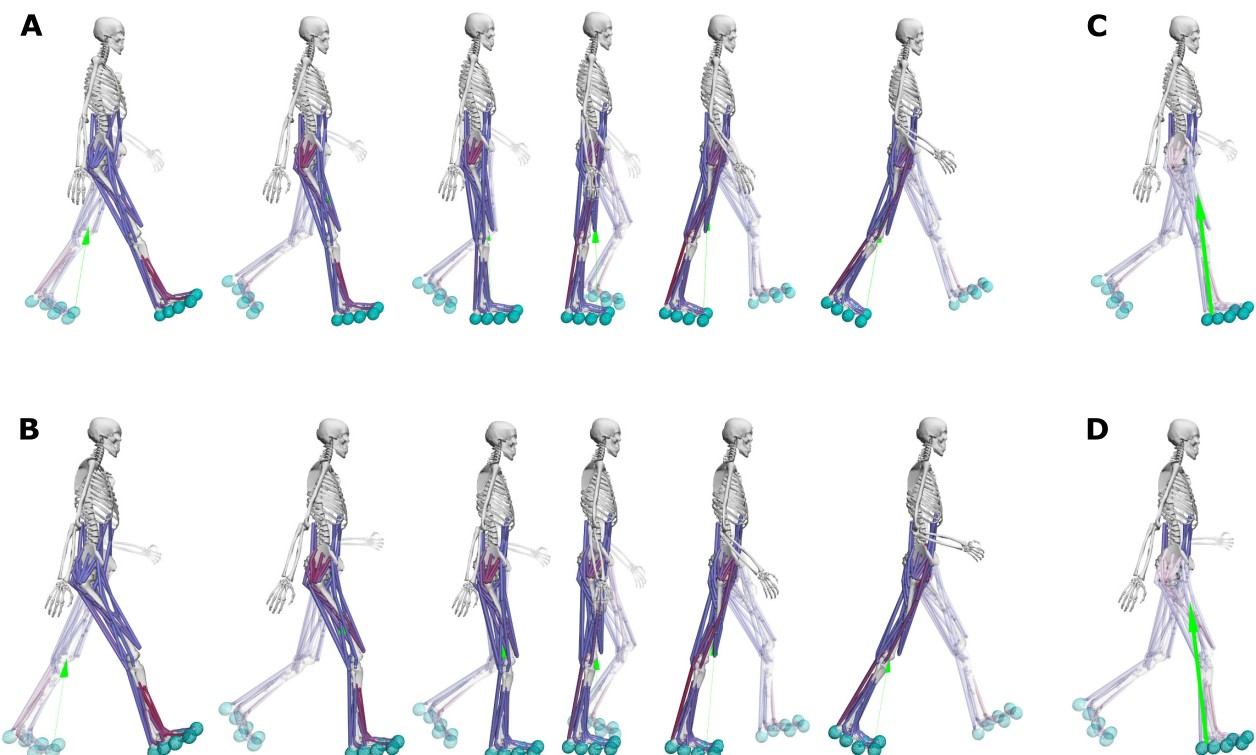

**Fig 2. Snapshots of the predicted half gait cycles.** Predicted half gait cycle with the old model with low contact spheres and without toes (**A**; solid black curve in Fig 1) and with the new model with high contact spheres and with toes (**B**; dashdot orange curve in Fig 1). We can appreciate the difference in knee flexion during early stance. The ground reaction force vector passes close to the knee joint center during early stance for the model without toes (**C**), whereas it is posterior to the knee joint center for the model with toes (**D**), requiring knee extension torques.

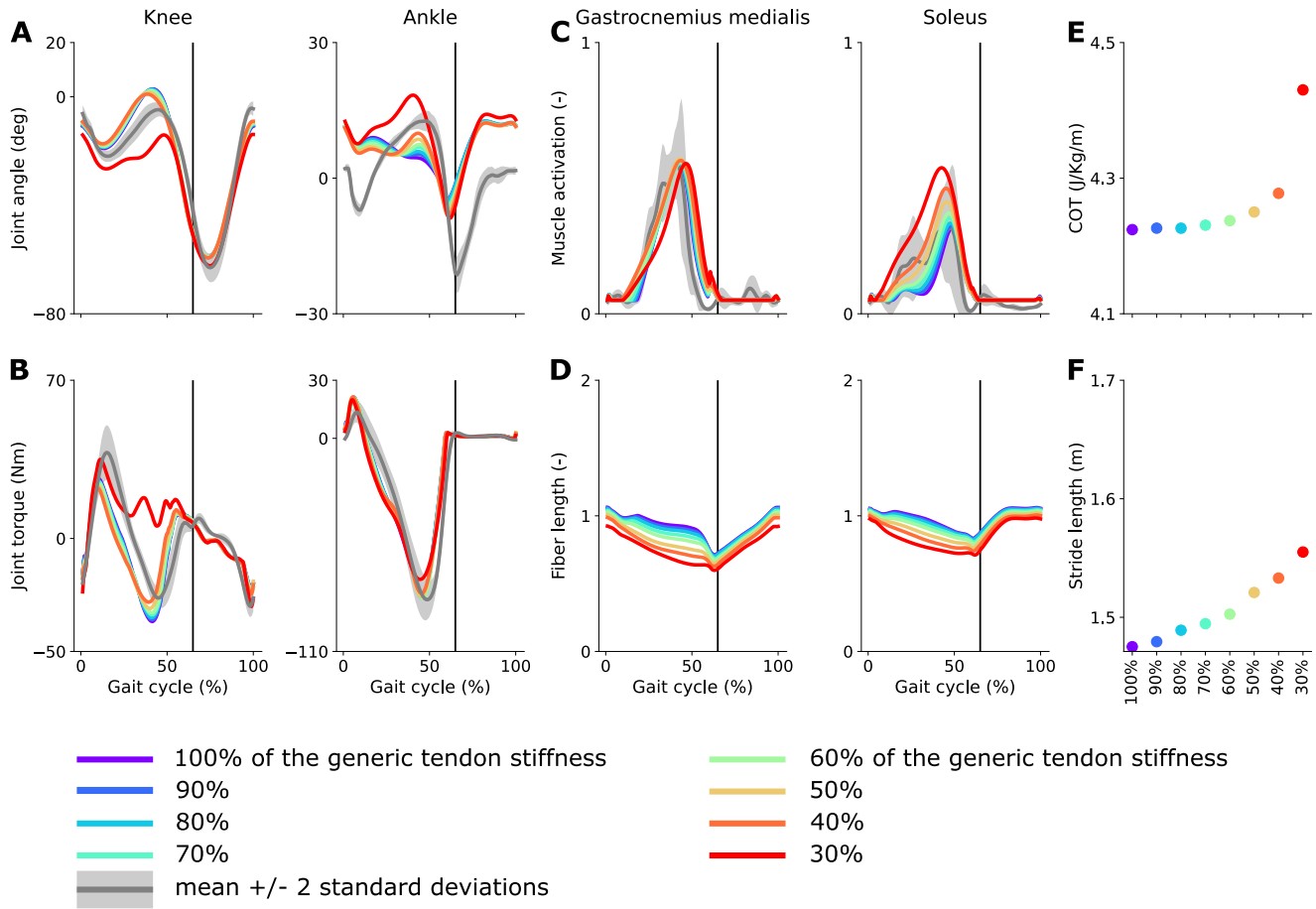

**Fig 3. Influence of Achilles tendon stiffness on predicted gait pattern.** Predicted knee and ankle kinematics (**A**) and kinetics (**B**), triceps surae activations (**C**), and normalized muscle fiber lengths (**D**) with models with different Achilles tendon stiffness. Predicted metabolic cost of transport (**E**) and stride length (**F**) as a function of the Achilles tendon stiffness (in percent of the nominal value). Experimental data (shaded areas) are shown as mean ± 2 standard deviations. The experimental electromyography data were normalized to peak nominal activations (100% of the generic Achilles tendon stiffness). The vertical black lines indicate experimental stance to swing transition.

new model with a lighter torso and heavier legs, with high contact spheres, and with toe joints (Fig 1).

## Influence of the Achilles tendon stiffness

Decreasing Achilles tendon stiffness by up to 60% improved stance ankle kinematics (Fig 3), with largest improvements with a 60% decrease (RMSE decreased from 8.2 to 6.3 Nm [-6.1%]). Decreasing Achilles tendon stiffness by 60% barely affected knee kinematics, but slightly worsened knee and ankle kinetics (RMSE increased from 10.6 to 13.2 [4%] and from 6.2 to 8.1 [2%], respectively). Decreasing Achilles tendon stiffness by more than 60% had a large effect on the knee and ankle kinematics and kinetics during stance. Soleus activity increased and turned on earlier with decreasing tendon stiffness. The gastrocnemius activity barely changed when adjusting tendon stiffness. For both muscles, the normalized fiber length profile was only slightly affected by the change in tendon stiffness, although both muscles operated at lower normalized fiber lengths with decreasing tendon stiffness. Metabolic cost of transport and stride length increased with decreasing tendon stiffness.

## Discussion

Musculoskeletal modeling choices can have a large influence on predictive simulations of walking. Here, we investigated the effect of two out of many possible modeling parameters, namely the number of degrees of freedom in the foot and the stiffness of the Achilles tendon, inspired by two common shortcomings of predictive simulations: a lack of stance knee flexion and poor stance ankle kinematics. These gait features likely result from the complex interplay between motor control and musculoskeletal dynamics. While previous studies have focused on the effect of the performance criterion on the predicted walking pattern [4, 5], our study reveals that a more physiologically plausible model of the feet, which incorporate toe joints, contributes to robustly eliciting knee flexion during stance but also improves stance ankle kinetics and reduces overestimation of the first vertical ground reaction force peak. Adding toe joints slightly worsened predictions of ankle kinematics, which were already of limited accuracy when using a model without toe joints. Decreasing Achilles tendon stiffness by up to 60% slightly improved the realism of the simulated ankle kinematics, but further reduction drastically altered the simulated gait pattern. We argue that the causal relationships between musculoskeletal properties and gait patterns that this study revealed are not intuitive, underlining the power of computer simulations to shed light on the role of isolated neuro-musculoskeletal features in shaping walking patterns. Yet, given that many combinations of performance criteria and musculoskeletal properties might generate the same gait features, experimental validation remains critical to ascertain that simulations are physiologically plausible. While it is clear that humans have toes, determining joint impedance, muscle-tendon properties and geometry experimentally is more challenging.

Models of foot-ground interaction shape how ground reaction forces are transmitted through the joints, which might explain why adding toe joints had a large influence on the more proximal knee joints. We also found that the position of the contact spheres can affect gait features such as stance knee flexion angles and knee extension torques, which further supports the importance of foot-ground interaction models. In comparison to our previous simulation study [4], we adjusted the vertical position of the contact spheres such that the model better captured the experimentally observed distance between the ankles and the ground. This change in the ground contact model, together with the altered mass distribution over the segments, explains why the simulations based on the model without toes presented here have larger stance knee flexion angles and knee extension torques than the simulations we previously published [4]. However, when incorporating toes, the simulated gait patterns were robust against these changes in contact location and mass distribution (Fig 1). This agrees with our expectation that subjects wearing shoes with thicker soles and having a heavier trunk would still walk with stance knee flexion. The current foot and foot-ground contact models still overly simplify the complex human foot and foot-ground interactions. Simulations could help further elicit the role of foot dynamics in shaping walking patterns.

Although our results suggest that toes contribute to eliciting stance knee flexion, motor control objectives might also shape stance knee flexion. Previous simulation studies have demonstrated that the performance criterion influences stance knee flexion (e.g., [5]). Yet the relationship between the performance criterion and stance knee flexion has not been unambiguous, suggesting that other factors play a role as well. Other performance criteria, such as reducing impact and improving stability [8], have also been suggested to induce stance knee flexion but, as far as we know, these hypotheses have not been tested either in simulation or through experiments. While it would be technically straightforward to penalize impact in the cost function when performing predictive simulations, it is more challenging to incorporate stability constraints. Simulations can be considered stable when they are robust against internal and external perturbations. Stability can be obtained through increased impedance by

coactivation of antagonist muscles [33, 34] and/or feedback control to react to unexpected perturbations, which would require solving stochastic optimal feedback simulations. Whereas such robust simulations have been generated for simple models [35], it remains challenging to perform robust simulations based on complex models. Predictive simulations of walking relying on reflex-driven control models have been shown to be robust against perturbations [36]. Yet predicting realistic knee mechanics has also been challenging with such models [6, 37, 38], possibly because the pre-imposed reflex structure does not sufficiently capture human motor control or because of model simplifications such as the absence of toe joints. In this study, we used the same cost function for all simulations, and tested the influence of musculoskeletal modeling assumptions. Studying the interactions between mechanics and cost function was out of the scope of this work, but should be envisioned in future studies. Indeed, it is now difficult to draw general conclusions, since the effect of the mechanics might be different when using a different performance criterion. Furthermore, we used data and models scaled based on anthropometric data from only one subject. Future studies could include more subjects and evaluate whether simulations capture inter-subject variability, which could help elicit the origins of variability (e.g., differences in motor control objectives and mechanics).

We found that incorporating toe joints in the model increased the simulated cost of transport, suggesting that having toes does not improve locomotion efficiency. The increased metabolic energy expenditure in the knee extensors is in line with previous observations that knee flexion during stance is energetically costly [5]. Yet our simulations suggest that, with toes, a gait pattern requiring knee extension torques during stance optimizes performance, possibly because, as we found, it reduces the metabolic energy expenditure in the plantarflexors with respect to a straight knee pattern in the absence of toes. Song et al. also found that foot compliance increased metabolic cost taking a different approach. Instead of only adding toes, they also modeled compliance of the foot arch and plantar fascia and used reflex-driven 2D simulations of walking [39]. Our simulation results are in line with experimental observations during running where using shoe wear that increases foot stiffness reduces metabolic cost [40].

Our simulations with different Achilles tendon stiffness suggest that performance optimization might encourage muscles to work at relatively low contraction velocities. Such muscle behavior agrees with experimental studies showing that the fascicles of the gastrocnemius tend to act relatively isometrically during the stance phase of walking [41]. While decreasing Achilles tendon stiffness by more than 60% had a big influence on ankle and knee kinematics, plantarflexor muscle fiber lengths changed more gradually with tendon stiffness and ankle plantarflexor torques were little sensitive to tendon stiffness (Fig 3). We also found that passive damping in the toe joints interacted with Achilles tendon stiffness. When damping was reduced, smaller decreases in Achilles tendon stiffness (more than 20%) elicited large changes in kinematics (S4 Fig). Yet reducing damping had little effect on fascicle lengths. Even though reducing Achilles tendon stiffness improved stance ankle kinematics, differences with experimental data remained large, in particular at the beginning and end of the stance phase. A possible explanation for this observations is that the relationship between joint kinematics and plantarflexor muscle-tendon lengths is inaccurately modeled. Indeed, fiber length does not only depend on tendon stiffness but also on muscle-tendon length, which is a function of ankle kinematics. If we overestimate the decrease in muscle-tendon lengths with ankle plantarflexion, this would result in reduced ankle plantarflexion in predictive simulations where having nearly isometric fiber lengths seems to optimize performance. This hypothesis is supported by previous results from inverse analysis (i.e., simulations for which the kinematics are prescribed) comparing simulated fiber lengths to their experimental counterparts measured using ultrasound [16]. In that study, we found that when imposing joint kinematics and kinetics, the simulated change in gastrocnemius fiber length during stance exceeded the measured change. We obtained the best fits

between simulated and measured fiber lengths when reducing Achilles tendon stiffness by on average 60%, which is in agreement with the results from the predictive simulations of the present study. Note that fiber length tracking errors remained relatively high in those inverse simulations with reduced Achilles tendon stiffness. In addition, previous studies have reported a wide range of Achilles tendon moment arms, and moment arms determine how muscle-tendon lengths change with joint angles [42–44]. We therefore believe that more accurately modeling the foot and plantarflexor geometry might improve simulated ankle angles.

Our analysis of convergence suggests that the selected discretization (i.e., number of mesh intervals) is accurate enough, the selected convergence tolerance is strict enough, and the solutions are robust against different initial guesses. Overall, this suggests that the control and state trajectories were approximated with sufficient accuracy. Note that this does not mean that our results are not local optima. It is indeed possible that both straight and flexed knee patterns during stance are local optima and that musculoskeletal modeling assumptions make one local optimum more likely to be found that the other one. Our analysis of convergence also suggests the importance of comparing simulations from different initial guesses, since they may be completely different (e.g., hot-start vs cold-start in simulations with 50 mesh intervals; S1 Table). Ensuring that simulations are not impacted by using a finer mesh and a tighter convergence tolerance, or by the choice of the initial guess brings is important to gain confidence in the accuracy of the results. It was interesting, although surprising, to note that our simulations converged faster (i.e., with fewer iterations) from the cold-start than from the hot-start when using a high mesh density. This suggests a limited impact of using a hot-start for such simulations, but this observation should be confirmed in future studies before generalizing. Finally, while our analysis suggests that we could have run our simulations using a convergence tolerance of $10^{-4}$ and 50 mesh intervals per half gait cycle, we cannot ensure that such numerical choices will be valid when altering musculoskeletal mechanics or cost function, or when simulating different movements.

## Conclusion

Both mechanics and cost function shape simulations of human walking. While previous studies have focused on the role of the cost function, we here demonstrate the effect of mechanical assumptions on predictive simulations of walking. Incorporating toes in the musculoskeletal model contributed to robustly eliciting stance knee flexion, improved ankle and knee kinetics, and normalized the first vertical ground reaction force peak. Yet it did not improve stance ankle kinematics. In contrast, decreasing the stiffness of the Achilles tendon by up to 60% slightly improved stance ankle kinematics, but there remain large discrepancies with experimental data. The poor predictions of ankle kinematics in simulations of walking thereby remain an open question, and further work is required to better understand how mechanical factors shape human walking. Computer simulations provide a useful tool to investigate the effect of the mechanics on human movements. However, as model complexity increases, so do the interactions between modeling assumptions, making it more difficult to comprehensively assess the effect of model parameters. Validating through experiments the effect of modeling assumptions observed in simulations would increase confidence in simulation outcomes, contributing to their deployment for applications such as personalized medicine and assistive device design.

## Supporting information

**S1 Appendix. Effect of the toe joint damping value on the predicted walking pattern and details about the smooth approximation of the metabolic energy model.**
(DOCX)

**S1 Fig. Predicted joint kinematics.** Experimental data (shaded areas) are shown as mean ± 2 standard deviations. The vertical black lines indicate experimental stance to swing transition. The experimental data for the pelvis tz (lateral pelvis displacement) are not shown due to large variations caused by different walking directions followed by the subject during the data collection. The old and new models have the same mass but different mass distribution: the new model having a lighter torso but heavier legs as compared to the old model.
(EPS)

**S2 Fig. Predicted joint kinetics.** Experimental data (shaded areas) are shown as mean ± 2 standard deviations. The vertical black lines indicate experimental stance to swing transition. The experimental data for subtalar torques are not shown given the large variability, which we suspect is due to measurement and modeling errors in the inverse dynamic analysis rather than representing true variability.
(EPS)

**S3 Fig. Predicted muscle activations.** Experimental data (shaded areas) are shown as mean ± 2 standard deviations. The vertical black lines indicate experimental stance to swing transition. The experimental electromyography data were normalized to peak activations of the new model with high contact spheres and with toes (dashdot orange curve).
(EPS)

**S4 Fig. Influence of Achilles tendon stiffness on predicted gait pattern with lower toe joint damping value (0.5 instead of 2 Nm s rad$^{-1}$).** Predicted knee and ankle kinematics (**A**) and kinetics (**B**), triceps surae activations (**C**), and normalized muscle fiber lengths (**D**) with models with different Achilles tendon stiffness. Predicted metabolic cost of transport (**E**) and stride length (**F**) as a function of the Achilles tendon stiffness (in percent of the nominal value). The experimental electromyography data were normalized to peak nominal activations (100% of the generic Achilles tendon stiffness). The vertical black lines indicate experimental stance to swing transition.
(EPS)

**S1 Table. Influence of the mesh density on the convergence profile.**
(DOCX)

**S2 Table. Influence of the convergence tolerance on the convergence profile.**
(DOCX)

**S1 Movie. Predictive simulations of walking with the old model (heavier torso but lighter legs) with low contact spheres and without toe joints.** Muscles turn red when active. The green arrows represent the ground reaction forces. The playback speed is 0.2 times real-time.
(MP4)

**S2 Movie. Predictive simulations of walking with the new model (lighter torso but heavier legs) with high contact spheres and with toe joints.** Muscles turn red when active. The green arrows represent the ground reaction forces. The playback speed is 0.2 times real-time.
(MP4)

## Acknowledgments

The authors would like to thank Tom Van Wouwe for helping with data collection.

## Author Contributions

**Conceptualization:** Antoine Falisse, Maarten Afschrift, Friedl De Groote.

**Data curation:** Antoine Falisse.

**Formal analysis:** Antoine Falisse, Friedl De Groote.

**Funding acquisition:** Friedl De Groote.

**Investigation:** Antoine Falisse, Maarten Afschrift, Friedl De Groote.

**Methodology:** Antoine Falisse, Maarten Afschrift, Friedl De Groote.

**Project administration:** Friedl De Groote.

**Resources:** Antoine Falisse, Maarten Afschrift.

**Software:** Antoine Falisse, Maarten Afschrift.

**Supervision:** Friedl De Groote.

**Validation:** Antoine Falisse, Maarten Afschrift, Friedl De Groote.

**Visualization:** Antoine Falisse, Maarten Afschrift, Friedl De Groote.

**Writing – original draft:** Antoine Falisse, Maarten Afschrift, Friedl De Groote.

**Writing – review & editing:** Antoine Falisse, Maarten Afschrift, Friedl De Groote.

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
