## [Decision Letter · Decision Letter 0]

13 Oct 2021

PONE-D-21-24376Modeling toes contributes to realistic stance knee mechanics in three-dimensional predictive simulations of walkingPLOS ONE

Dear Dr. Falisse,

Thank you for submitting your manuscript to PLOS ONE. After careful consideration, we feel that it has merit but does not fully meet PLOS ONE’s publication criteria as it currently stands. Therefore, we invite you to submit a revised version of the manuscript that addresses the points raised during the review process.

The reviewers raised serios concerns about model justification, methods of analysis, and significance of findings. Please, revised the manuscript to address in detail the following critique and questions raised by the reviewers:

Authors added the toes and metatarsophalangeal joints. Authors claim that this modification “contributed to robustly eliciting physiological stance knee flexion angles, knee extension torques, and knee extensor activity.” However, at the same time ankle joint angle became worse at the middle of gait (see S1Fig). It is not clear why reproducing better knee joint angle is more important than ankle joint angle.

Please, justify using experimental data of just one subject.  

Please, provide data on number of recorded experimental cycles.

On figures, experimental data represented only by shaded area without mean pattern. On some panels it is impossible to see how experimental patterns look like. For example, subtalar angle on S2Fig. There is no information where stance and swing phases on figures.

   The introduction does not provide enough justification of why specifically it is important to investigate the influence of foot complexity and Achilles tendon on walking, with the latter raising even more questions when it does not yield a positive result. The lack of ankle plantarflexion and knee flexion are mentioned, but it is unclear why these features of human gait are important. Is there any evidence that the lack of them leads to instability or them being present is a symptom?

   Many of the statements about differences between models are made based on qualitative evaluation of the figures, without quantification. In most cases, it is not very visible on plots with 5 lines being very close to each other, and can be supported simply with a correlation, which will provide an exact value for the lack of difference or presence of small differences.

  The Results related to optimization evaluation, e.g., mesh density and tolerance, are very important for model validation and reproduction, but are not generalizable, as authors specify themselves, and would be more appropriate for Methods.

Minor:

L47-49: This needs a citation or a description of the effect.

L50: Could you describe what is the terminal stance exactly? Including a description of the major states within step cycle might help a novice reader navigate.

L71-74: A very long sentence, consider splitting.

L117: How much of a change it is from the generic model? Is there anatomical evidence to support either of these orientations?

L122: How were these specific values chosen?

L138: What technology was used to record 3D marker positions, which devices were used for EMG recordings, how were ground reaction forces recorded?

L145: How many and which muscles were recorded?

L145: The EMG was first filtered on two non-overlapping bands: first 20-400 Hz, and then <10Hz. There should be minimal signal left. Are there examples of the prior use of this method?

L160: This section should provide a specific formula for the metabolic energy rate, as well as the quality of approximation.

L180: Why was the generic torso mass changed?

L220-228: These statements should be supported by quantified measurements that specify how much the model results differ, e.g., a correlation between traces, possibly in Methods.

Figures 1 and 2 are in the wrong order.

Figure 2A legend would benefit from specifying shaded region as experimental data.

Table 1. Would benefit from including the experimental metabolic cost for comparison.

L246-252: Same for quantification of changes.

The result that the vertical GRF has smaller peak in models with toes should be highlighted more as a benefit of toe model, although leading to higher metabolic cost. It seems to be important enough to be highlighted in the discussion, conclusion and abstract.

What was the main contributor to the increased metabolic cost?

L272-282: It would be interesting to see which values of the generic tendon stiffness are closer to the physiological behavior curves.

What is the scientific purpose of mesh density and tolerance measurements? Can this be moved to Methods?

L357: Are there any comparative studies supporting this?

L371: Stability can also be produced by coactivation of antagonistic muscles, which act in anticipation of a perturbation. This should be mentioned, e.g. with references to Hogan 1984 in IEEE, Stroeve 1999 in Bio Cybernetics.

Source code contains models for two subjects: https://github.com/antoinefalisse/predictsim_mtp/tree/master/OpenSimModel

Could you specify which one was used?

We look forward to receiving your revised manuscript.

Kind regards,

Gennady S. Cymbalyuk, Ph.D.

Academic Editor

PLOS ONE

Journal Requirements:

Reviewers' comments:

Reviewer's Responses to Questions

**Comments to the Author**

1. Is the manuscript technically sound, and do the data support the conclusions?

Reviewer #1: Yes

Reviewer #2: Partly

2. Has the statistical analysis been performed appropriately and rigorously? 

Reviewer #1: No

Reviewer #2: N/A

3. Have the authors made all data underlying the findings in their manuscript fully available?

Reviewer #1: Yes

Reviewer #2: Yes

4. Is the manuscript presented in an intelligible fashion and written in standard English?

Reviewer #1: Yes

Reviewer #2: Yes

5. Review Comments to the Author

Reviewer #1: This manuscript describes a development of a musculoskeletal model of human lower body. The authors altered the mass distribution, skeletal and muscular properties of the foot and investigated their influence on the modelled human walk. This is a technical report of good quality, however, there are a several issues that need to be addressed before publication.

Major:

1. The introduction does not provide enough justification of why specifically it is important to investigate the influence of foot complexity and Achilles tendon on walking, with the latter raising even more questions when it does not yield a positive result. The lack of ankle plantarflexion and knee flexion are mentioned, but it is unclear why these features of human gait are important. Is there any evidence that the lack of them leads to instability or them being present is a symptom?

2. Many of the statements about differences between models are made based on qualitative evaluation of the figures, without quantification. In most cases, it is not very visible on plots with 5 lines being very close to each other, and can be supported simply with a correlation, which will provide an exact value for the lack of difference or presence of small differences.

3. The Results related to optimization evaluation, e.g., mesh density and tolerance, are very important for model validation and reproduction, but are not generalizable, as authors specify themselves, and would be more appropriate for Methods.

Minor:

L47-49: This needs a citation or a description of the effect.

L50: Could you describe what is the terminal stance exactly? Including a description of the major states within step cycle might help a novice reader navigate.

L71-74: A very long sentence, consider splitting.

L117: How much of a change it is from the generic model? Is there anatomical evidence to support either of these orientations?

L122: How were these specific values chosen?

L138: What technology was used to record 3D marker positions, which devices were used for EMG recordings, how were ground reaction forces recorded?

L145: How many and which muscles were recorded?

L145: The EMG was first filtered on two non-overlapping bands: first 20-400 Hz, and then <10Hz. There should be minimal signal left. Are there examples of the prior use of this method?

L160: This section should provide a specific formula for the metabolic energy rate, as well as the quality of approximation.

L180: Why was the generic torso mass changed?

L220-228: These statements should be supported by quantified measurements that specify how much the model results differ, e.g., a correlation between traces, possibly in Methods.

Figures 1 and 2 are in the wrong order.

Figure 2A legend would benefit from specifying shaded region as experimental data.

Table 1. Would benefit from including the experimental metabolic cost for comparison.

L246-252: Same for quantification of changes.

The result that the vertical GRF has smaller peak in models with toes should be highlighted more as a benefit of toe model, although leading to higher metabolic cost. It seems to be important enough to be highlighted in the discussion, conclusion and abstract.

What was the main contributor to the increased metabolic cost?

L272-282: It would be interesting to see which values of the generic tendon stiffness are closer to the physiological behavior curves.

What is the scientific purpose of mesh density and tolerance measurements? Can this be moved to Methods?

L357: Are there any comparative studies supporting this?

L371: Stability can also be produced by coactivation of antagonistic muscles, which act in anticipation of a perturbation. This should be mentioned, e.g. with references to Hogan 1984 in IEEE, Stroeve 1999 in Bio Cybernetics.

Source code contains models for two subjects: https://github.com/antoinefalisse/predictsim_mtp/tree/master/OpenSimModel

Could you specify which one was used?

Reviewer #2: This work is slightly modified published simulation work by the same authors. Authors added the toes and metatarsophalangeal joints. Authors claim that this modification “contributed to robustly eliciting physiological stance knee flexion angles, knee extension torques, and knee extensor activity.” However, at the same time ankle joint angle became worse at the middle of gait (see S1Fig). It is not clear why reproducing better knee joint angle is more important than ankle joint angle. Authors compared their simulation with experimental data of one subject. There is no information of number of recorded experimental cycles. On figures, experimental data represented only by shaded area without mean pattern. On some panels it is impossible to see how experimental patterns look like. For example, subtalar angle on S2Fig. There is no information where stance and swing phases on figures.

6. PLOS authors have the option to publish the peer review history of their article (what does this mean?). If published, this will include your full peer review and any attached files.

Reviewer #1: No

Reviewer #2: No

---

## [Author Response · Author response to Decision Letter 0]

8 Dec 2021

Dear Editor and reviewers,

Many thanks for giving us a chance to submit a revised version of our manuscript. We have addressed all comments of the reviewers. Some of our results slightly changed when re-running simulations to answer some of the reviewers’ questions. We found that using a higher toe joint damping value produced simulations that better reproduced experimental data. In the revised version of this manuscript, we therefore present slightly updated results, and we adjusted our statements in different parts of the manuscript to reflect our new results. We believe our paper improved by a lot with this round of review and would like to thank the reviewers for all the comments they made. We hope the paper now satisfies the standards of the journal, but we would be happy to further clarify certain parts if deemed necessary. We apologize for the delay of the revision, but the server of PLOS ONE was down for the last two weeks and we were unable to submit our revision.

You will find our answers to the reviewers’ questions in the document Response to Reviewers. The answers are in blue and the modified text in red.

Yours sincerely,

Antoine Falisse on behalf of all co-authors.

---

## [Decision Letter · Decision Letter 1]

3 Jan 2022

PONE-D-21-24376R1Modeling toes contributes to realistic stance knee mechanics in three-dimensional predictive simulations of walkingPLOS ONE

Dear Dr. Falisse,

Thank you for submitting your manuscript to PLOS ONE. After careful consideration, we feel that it has merit but does not fully meet PLOS ONE’s publication criteria as it currently stands. Therefore, we invite you to submit a revised version of the manuscript that addresses the points raised during the review process. Please, list in the methods the muscles, which have been evaluated in the manuscript. Since the used approximation of the original formula from Bhargava et al., 2004  has not been described elsewhere, it should be provided in the manuscript for reproducibility. Please submit your revised manuscript by Feb 17 2022 11:59PM. If you will need more time than this to complete your revisions, please reply to this message or contact the journal office at plosone@plos.org. Please include the following items when submitting your revised manuscript:A rebuttal letter that responds to each point raised by the academic editor and reviewer(s). You should upload this letter as a separate file labeled 'Response to Reviewers'.A marked-up copy of your manuscript that highlights changes made to the original version. You should upload this as a separate file labeled 'Revised Manuscript with Track Changes'.An unmarked version of your revised paper without tracked changes. You should upload this as a separate file labeled 'Manuscript'.If applicable, we recommend that you deposit your laboratory protocols in protocols.io to enhance the reproducibility of your results. Protocols.io assigns your protocol its own identifier (DOI) so that it can be cited independently in the future. For instructions see: https://journals.plos.org/plosone/s/submission-guidelines#loc-laboratory-protocols. Additionally, PLOS ONE offers an option for publishing peer-reviewed Lab Protocol articles, which describe protocols hosted on protocols.io. Read more information on sharing protocols at https://plos.org/protocols?utm_medium=editorial-email&utm_source=authorletters&utm_campaign=protocols.

We look forward to receiving your revised manuscript.

Kind regards,

Gennady S. Cymbalyuk, Ph.D.

Academic Editor

PLOS ONE

Journal Requirements:

Reviewers' comments:

Reviewer's Responses to Questions

**Comments to the Author**

1. If the authors have adequately addressed your comments raised in a previous round of review and you feel that this manuscript is now acceptable for publication, you may indicate that here to bypass the “Comments to the Author” section, enter your conflict of interest statement in the “Confidential to Editor” section, and submit your "Accept" recommendation.

Reviewer #1: All comments have been addressed

Reviewer #2: All comments have been addressed

2. Is the manuscript technically sound, and do the data support the conclusions?

Reviewer #1: Yes

Reviewer #2: Yes

3. Has the statistical analysis been performed appropriately and rigorously? 

Reviewer #1: Yes

Reviewer #2: Yes

4. Have the authors made all data underlying the findings in their manuscript fully available?

Reviewer #1: Yes

Reviewer #2: Yes

5. Is the manuscript presented in an intelligible fashion and written in standard English?

Reviewer #1: Yes

Reviewer #2: Yes

6. Review Comments to the Author

Reviewer #1: The authors have addressed all the comments a critiques, greatly improving the manuscript. There are a couple minor points that be addressed.

“We specified the number of muscles in the revised version of the methods. We think listing all 24 muscles densify the text a lot while not adding much to the story. We therefore only indicated that those were 24 muscles in the lower limbs. Please note that electromyography was only used to evaluate the simulated muscle activations as now explicitly mentioned.” – The muscles are important to list in the methods because they identify the modelled muscles, which have been evaluated in the manuscript and which have been not. This may provide the directions for the future analysis of the model, e.g., which modelled muscle activity has not been validated.

“The formula for the metabolic energy rate is described in length in the cited publication [15], and we believe it is out of the scope of this paper to provide it again here, as it involves many terms with many variables that would need to be defined. Yet we revised this part of the methods to make clearer what terms were included in the computation of the metabolic energy rate.” – The comment from the first review was more to do with the approximation than the formula itself. Since not the original formula from Bhargava et al., 2004 is used, but an approximation of it, which is not described elsewhere, it should be provided in the manuscript for reproducibility, or the way it differs from the original.

Reviewer #2: All comments have been addressed. Authors added all necessary comments and explanations as well as modified figures as requested

7. PLOS authors have the option to publish the peer review history of their article (what does this mean?). If published, this will include your full peer review and any attached files.

Reviewer #1: No

Reviewer #2: No

---

## [Author Response · Author response to Decision Letter 1]

7 Jan 2022

Dear Editor,

Many thanks for giving us a chance to resubmit this manuscript. We have addressed the remaining comments. We hope the paper now satisfies the standards of the journal, but we would be happy to clarify certain parts if deemed necessary. You will find our answers to the reviewers’ questions in the Response To Reviewers document.

Yours sincerely,

Antoine Falisse on behalf of all co-authors.

---

## [Decision Letter · Decision Letter 2]

11 Jan 2022

Modeling toes contributes to realistic stance knee mechanics in three-dimensional predictive simulations of walking

PONE-D-21-24376R2

Dear Dr. Falisse,

We’re pleased to inform you that your manuscript has been judged scientifically suitable for publication and will be formally accepted for publication once it meets all outstanding technical requirements.

Kind regards,

Gennady S. Cymbalyuk, Ph.D.

Academic Editor

PLOS ONE

Additional Editor Comments (optional):

Reviewers' comments:

Reviewer's Responses to Questions

**Comments to the Author**

1. If the authors have adequately addressed your comments raised in a previous round of review and you feel that this manuscript is now acceptable for publication, you may indicate that here to bypass the “Comments to the Author” section, enter your conflict of interest statement in the “Confidential to Editor” section, and submit your "Accept" recommendation.

Reviewer #1: All comments have been addressed

2. Is the manuscript technically sound, and do the data support the conclusions?

Reviewer #1: Yes

3. Has the statistical analysis been performed appropriately and rigorously? 

Reviewer #1: Yes

4. Have the authors made all data underlying the findings in their manuscript fully available?

Reviewer #1: Yes

5. Is the manuscript presented in an intelligible fashion and written in standard English?

Reviewer #1: Yes

6. Review Comments to the Author

Reviewer #1: The authors have addressed all comments and all necessary additional information has been provided.

7. PLOS authors have the option to publish the peer review history of their article (what does this mean?). If published, this will include your full peer review and any attached files.

Reviewer #1: No

---

## [Editor Report · Acceptance letter]

14 Jan 2022

PONE-D-21-24376R2 

Modeling toes contributes to realistic stance knee mechanics in three-dimensional predictive simulations of walking 

Dear Dr. Falisse:

I'm pleased to inform you that your manuscript has been deemed suitable for publication in PLOS ONE. Congratulations! Your manuscript is now with our production department. 

Kind regards, 

on behalf of

Dr. Gennady S. Cymbalyuk 

Academic Editor

PLOS ONE